# Challenges in the Valorization of the Funerary Heritage; Experiences in the Municipal Cemetery of Murcia (Spain)

Gabriel López-Martínez [1,*] and Klaus Schriewer [2]

1 Department of Contemporary Humanities, University of Alicante, 03690 Alicante, Spain
2 Center for European Studies (CEEUM), Faculty of Philosophy, University of Murcia, 30100 Murcia, Spain; ks@um.es
* Correspondence: gabriel.lopez@ua.es

**Abstract:** The cemetery is a cultural landscape that represents themes of great relevance to interpret the structure of a society, roles, and hierarchies, as a reflection of its social life. The cemetery gathers a whole symbolic universe where local social histories are represented, beyond the history of art and the architectural aspect. As a heritage element, the cemetery shows us the socio-cultural changes of a territory: religious questioning, changes linked to the family, individualization of contemporary society or broader questions about socio-economic structure. This article presents the experience conducted during the last 6 years in the Cemetery "Nuestro Padre Jesús" in Murcia (Spain), through a collaboration among the Sociedad Murciana de Antropolgía (SOMA), the University of Murcia and the Municipality of Murcia, developing the project "Funerary Cultures", whose main objective is to promote the heritage, cultural and historical values of the funerary culture. Specifically, as a result of this teaching innovation experience, the six thematic guides to visit the cemetery are presented as an experience of *patrimonialization* of elements of the cemetery and its consequent selection and consensus exercise to determine what was considered as heritage in the context of the cemetery. Finally, a proposal of a systematic process in the valuation and selection of the material objects in the cemetery is presented; this proposal allows us to establish a debate on what considerations to take into account when considering the relationship between cultural heritage and the cemetery as a cultural landscape in permanent transformation.

**Keywords:** cemetery; cultural heritage; funerary heritage; funerary cultures

## 1. Introduction

Human beings' attitude towards death has always been a dialogue with their own time. The cemetery—the predilect place where this dialogue materializes—is a frontier, the space in which individuals, the human body, finds its limit and, therefore, a place in which to pose the meaning that death has for life [1,2]. As an imagined and constructed enclosure, the cemetery allows us to understand how the city reinterprets its relationship with those who inhabited it, and what its function and meaning is. In this sense, the cemetery is the territory of remembrance and collective memory. It fulfills a social function, a double task of a hygienic and symbolic nature. On the one hand, it responds to the need to regulate the separation between the living and the bodies of the deceased, and, on the other, it has a cultural and symbolic heritage dimension. In this second function, art, the epitaphs engraved in stone, with a claim to permanence, have their place However, with the same importance, the cemetery is a reflection of historical periods and events, of the social hierarchy, and can be analyzed as a mirror of a society and its history.

In the 19th century, Spanish cemeteries had a monumental character, which they have been losing throughout the 20th century, and whose evolution continues today, with cremation gaining ground over burial. In other European places, cemeteries of "romantic heritage" were developed, conceived as places for strolling and reflection; the best-known

example would be the Père-Lachaise in Paris. Even today, a novel successor to the Romantic cemetery, the so-called Friedwald (Forest of Peace), is being developed in Europe, which allows one to establish a dialogue between death and nature. On the other hand, in our country and particularly in the Mediterranean area, a type of cemetery has developed to be as what we describe as architectural, with very close constructions and narrow alleys.

In recent decades in Europe and especially in the north of the subcontinent, the heritage value of cemeteries has been (re)discovered. There is growing anthropological and historical research on these sites, which are presented as reflections of local histories that foster the perception of cemeteries as heritage. Linked to this process is the growing tourist use that can be seen among others in the creation of the Association of Significant Cemeteries in Europe (ASCE).

In addition to appreciating the artistic and architectural value of certain elements of cemeteries, a much broader reflection is needed in order to avoid certain cultural prejudices (i.e., it is necessary to clearly point out what other assets and values of a cultural nature need to be protected in cemeteries). What kind of tombs are worth preserving: only those that have a certain aesthetic value or also those that provide cultural significance and, therefore, bear witness and testimony to time? The concept of cultural heritage linked to cemeteries invites us to point out a great variety of singularities and identities that need protection: the relevant personalities in the history of the city, the historical testimonies and, among them, that of the Spanish Civil War, and also the specific areas that were traditionally reserved for children, the non-baptized, non-Catholic Christians and, in general, the so-called dissidents; those "other" deceased dissidents; those somewhat invisible "other" deceased who, nevertheless, shared life and death with the great Catholic majority. Therefore, the answer we give to these questions implies the creation and legitimization of what we consider to be our "cultural heritage" and which, consequently, we judge and respect as a social good that deserves to be studied, preserved and transmitted to future generations.

Since 2015, the "Sociedad Murciana de Antropología" (SOMA), in collaboration with the City Hall of Murcia, has been developing the project "Funerary Cultures", whose main objective is to promote the heritage, cultural, and historical values of the funerary culture. We start from the hypothesis that the cemetery is a privileged place to tell the history and socio-cultural processes experienced by any society. The specific place of this work is the municipal cemetery of the city of Murcia, "Nuestro Padre Jesús" (Murcia, Spain). The activities conducted in the "Funerary Cultures" project are divided into different areas. On the one hand, thematic visitor guides are being prepared and published annually. On the other hand, guided tours of the cemetery are offered. The visitor guides are the result of a teaching innovation project conducted in the Faculty of Philosophy of the University of Murcia, and consists of a study and research by students, whose product are twelve annual biographies that are presented in each visitor guide and that provide certain elements or a new perspective in the story of the history of Murcia. Between 2016 and 2021, we have published six visitor guides. Each one has been dedicated in a monographic way to a theme: writers and artists, elites of the nineteenth century, women in memory, protagonists of progress, European immigrants and, the last guide produced, the 1918 flu epidemic in Murcia.

In this article, we intend to describe and analyze the experience of cooperation between SOMA, the University of Murcia and the Municipality of Murcia, and to present a proposal for a systematic procedure for the selection and valuation of the elements housed in the cemetery. For this purpose, we will describe in the first part the beginnings of the cooperation and the different activities conducted. In the second section, we will present the current challenges that the declaration of the cemetery as an asset of cultural interest entails. Finally, in the third section we will present a proposal of how a systematic process could be established in the valuation and selection of the material objects in the cemetery. It is a reflection of this experience conducted in recent years, which allows us to establish a debate on what considerations to take into account when considering the relationship between cultural heritage and the cemetery as a cultural landscape in permanent transformation.

Beyond this, we intend to reflect on the criteria that allow us to talk about the consideration of the cemetery as an Asset of Cultural Interest.

Before offering this analysis of the experience conducted, the following section presents a bibliographical review of the origins of the contemporary cemetery, the concept of cultural heritage and, finally, the idea of the cemetery as cultural heritage.

## 2. Theoretical Framework

The historiography of the cemetery has been dominated by two main theoretical frameworks [3]. On the one hand, Marxist characterizations of class struggle and the display of status in the cemetery landscape [4,5]. On the other, and more recently, the articulation of the nature of Foucauldian "biopolitics" and governance expressed through the bureaucratization of modes of burial and sanitary technologies of disposal [6–8]. These two areas would occupy a theoretical field of reflection, asking about the origins, the ontology of the cemetery or socio-political aspects in this context. In our case, by presenting a case of practical experience, in this section we will offer a review of (a) the origins of the contemporary cemetery; (b) international evolution and considerations on the concept of cultural heritage; and (c) considerations on the cemetery as cultural heritage.

### 2.1. Origins of the Contemporary Cemetery

For centuries, the term "cemetery" has been a generic reference to a delimited burial place, whether it was the space surrounding a church or a cemetery outside the walls. Until the Modern Age, death and burial were the domain of the Church. However, the history of burial becomes an example of the secularization project of the Enlightenment, so that the deceased body is removed from the spiritual realm of the holy field and placed in the scientific realm of the cemetery [9].

In the 16th century, hygienic criticisms of the overcrowded urban church cemeteries, on the one hand, and the religious and social resurgence of reform movements, on the other, heralded the gradual end of burials in church cemeteries. From then on, cemeteries were built outside the cities, but still under the ownership of the Church. As a consequence of the Enlightenment, the Reformation and population growth, cemeteries were relocated again in the eighteenth and nineteenth centuries [10–12]. Therefore, we may consider that the contemporary cemetery emerged in the 18th century as a device to solve a sanitary problem. The hygienist movement, which developed throughout Europe, emphasized the state's obligation to organize the health of the city and its inhabitants, which prompted important changes in mentality based on medical reasons [13].

France and Austria offer some important examples. Towards the middle of the 18th century, the French capital criticized burial in mass graves and promoted the creation of individual tombs. Criticism arose mainly over the Cemetery of the Innocents, located within the city of Paris. A total of one tenth of the deceased Parisians were buried there, which had led to catastrophic hygienic conditions. A royal declaration issued in 1776 called for the relocation outside the walls of urban cemeteries throughout the country that posed a health hazard and further resulted in the prohibition, in 1780, of all burials in the Parisian Cemetery of the Innocents (which was subsequently excavated in 1785–87). French reform efforts were provisionally ended by the décret impérial sur les sepultures, promulgated by Napoleon I in 1804. Through the legislation of the occupied French territories, the Napoleonic decree had a direct impact on other states in the early 19th century and caused a wave of cemetery relocations, the most famous example being that of the Père Lachaise in Paris, founded in 1804, and also known as The Cemetery of Eastern Paris [14].

Throughout the 19th century, the aesthetic ideal for the structuring of cemeteries was progressively oriented towards the English garden style. Thus, in the course of the 19th century, municipal cemeteries had become a place for strolling. The *aesthetization* of the landscape of European cemeteries had reached a temporary peak in the cemetery parks, whose natural landscape served as a backdrop for an increasingly overflowing cult of the sepulcher, especially at the end of the 19th century. It was also a very important cultural and

sociological change. The tombs truly celebrated the bourgeois individual: factory owners, teachers, civil servants, all wanted their biographies immortalized on their tombstones. The image of death was not only emotionally charged, but also personalized. Portraits of the deceased, such as reliefs, were frequent. In addition, the style of the tombs became more and more diversified. In particular, the historicism of the late 19th century gave rise to an exuberant stylistic flowering: neo-baroque forms coexisted with neo-gothic and neoclassical ones. Towards 1900, the tombs became increasingly monumental. The culmination was the cult of the mausoleum: mausoleums were considered particularly aristocratic forms of sepulchre; they were expensive and reserved for a reduced social elite. The tombs of the 19th century thus reflected heritage, education, and social prestige in a multitude of forms: the triumphant trajectory of the bourgeoisie that contributed to consolidating its identity [12].

From the end of the 19th century, the introduction of modern cremation transformed cemeteries once again. It appeared in the context of industrialization, technification, and urbanization. Burial was accelerated by the cremation process. The burial of ashes on the one hand reduced the space required in cemeteries and on the other hand allowed diversification of burial possibilities beyond the cemetery. In this sense, the introduction of modern cremation brought about a fundamental reform of funerary culture. In the reformist era of the 1920s, the cinerary urn became an exemplary model for the standardization and unification of funerary culture. At the same time, the transition to communal urn cemeteries and anonymous urn fields with virtually no individual stelae, which followed the Second World War, was already in the offing [15].

Our experience proposes to expand the social use of the cemetery as an activator of memories, a relational axis to understand the urban fabric, and an educational resource of great value, because from it we may elaborate new views and social practices about the cultural heritage of the city.

### 2.2. On the Concept of Cultural Heritage: Evolution and International Considerations

We can consider that the birth of the concept of Cultural Heritage, as we understand it today, is a social construction of the 20th century [16–18]. Before reaching this point, in the same manner so many other phenomena, the concept of cultural heritage was gestated in the revolutions and movements of the 19th century. In antiquity, the creations that have reached our days, have done so more by chance than by the desire for permanence, although it is possible to note some examples of goods that have been the object of esteem, which has favored their accumulation and preservation. In the Middle Ages, the Catholic Church and the monarchy were the pillars on which the artistic and architectural wealth of Europe is based, and the role of both institutions has been key to their conservation and also to their destruction. In the Renaissance, humanists will begin to look to classical antiquity as a model to follow. At this time, it could be said that the concept of cultural heritage begins to be introduced into history and art. We can already speak of a will to accumulate with a sense of memory and future. The outbreak of the French Revolution will provoke important changes in the world of culture. The people will demand access to artistic goods, as the property of the nation. It is at this moment when a whole series of goods that until then had been the property of the Church, kings, and nobles, became public property. Thus began the formulation of a concept of Cultural Heritage focused on the artistic and architectural aspects, monuments and works of art, quite distant, in general, from the daily life of the people.

It was not until the second half of the twentieth century that the notion of context took hold. It is an idea for which the place occupied and the function performed by the ancient object begins to have more importance than its value or its beauty, because it serves to explain the past much better than the latter. As a result of progress in the study of art, archaeology, history and anthropology, the concept of heritage will be redefined or rethought, broadening it and thus considering the need to conserve other types of property.

For a better understanding of the change from a heritage model that we can call ancient to a modern one, we can look at the following Table 1 [19].

**Table 1.** Differences between an ancient and a contemporary model of the concept of Cultural Heritage.

| Antique Model<br>Historical and Artistic Heritage | Contemporary Model<br>Cultural Heritage |
| --- | --- |
| **Restricted conception**. Criteria for selection and valorization based on the time factor (historical and archaeological witnesses) and artistic values and representativeness (scarcity and exceptionality). | **Open conception**. Manifestations of the cultural identities of different collectives through time. The identity of the present represents only the last phase of an unfinished process. The consideration of tradition or the traditional to delimit the sense of continuity of certain cultural components. |
| **Elitist**. Focused mainly on the most singular human creations, generally linked to the power elites. | Cultural creations that bear witness to the **lifestyles**, **values** and **beliefs** of the different social groups that comprise it and of society as a whole. |
| Limited to the production of movable and immovable goods made by human beings. | It encompasses both **tangible** and **intangible** culture as expressions of a people's ethnic identity and how it has been shaped over time. |
| Focused primarily on **material culture.** | It includes **natural assets** as assets to be valued and protected. Nature as territory and human beings are presented as inseparable realities. |

In this work, we consider cultural heritage as that which is constituted by a set of creations, of goods that a community or human group has been elaborating throughout its social and historical development, and that acquire a character of witness and reflection of its cultural identity. For these manifestations to be considered cultural heritage, they must be recognized, accepted, and assumed as heritage by that community.

Returning to the consideration of the social construction of heritage, we understand that in order to be effective and legitimized, the creation of heritage would consist of a constant struggle over the different considerations about what is heritage. In this sense, the experience offered in this paper is an example of the opposing perspectives when it comes to interpreting history through heritage [20,21]. That is to say, it must be placed in a certain social, political, and cultural context. In this construction of heritage, the concept of selection must necessarily be introduced. However, which principles and ideas guide selection? Selecting is a way of attributing value. The selection process is based on a set of framework values. These values depend on a given cultural, historical, and even psychological context. Depending on the context, some resources are more highly valued at a given time than others.

In the case of Spain, the protection of Cultural Heritage is linked to the Spanish Historical Heritage Law (1985). Its purpose is "the protection, enhancement and transmission to future generations of Spain's historical heritage, comprising movable and immovable property of artistic, paleontological, historical, archeological, ethnographic, scientific or technical interest, documentary and bibliographic heritage, archeological sites and areas, and natural sites, gardens and parks of artistic, historical or anthropological value". An important aspect of this law is the regime for the protection of historical heritage. The state law and its implemented regulations have established three levels of protection. The first level of protection is that of Spanish Historical Heritage. At a higher level of protection are the properties included in the General Inventory of Movable Property that have a notable historical, archaeological, scientific, technical, or cultural value, which have not been declared of cultural interest. Finally, the highest level of protection is constituted by movable and immovable property declared Objects or Assets of Cultural Interest (BIC).

The international dimension of cultural heritage is given by the growing influence of international organizations in cultural matters. The role of UNESCO has been, in this sense, of great influence. On the one hand, in the face of the globalizing process, UNESCO

has established one of the strongest institutional pillars of its discourse in defense of the diversity and cultural identity of local communities, including the drafting of the Convention for the Safeguarding of the Intangible Cultural Heritage (2003) or the Convention for the Protection and Promotion of the Diversity of Cultural Expressions (2005) [22,23]. However, at the same time, UNESCO has successfully promoted the definition of a World Heritage that, in this case, is linked to a heritage that transcends all national, regional, or local borders to be admired, protected, and enjoyed by the whole of humanity. In this respect, the cultural property declared World Heritage in a certain way becomes globalized by assuming this universal value and becoming part of humanity. All kinds of agents are involved in this process, from citizens' associations and platforms to the technical and bureaucratic apparatus of public administrations, in addition to the monitoring of the process by the media and not forgetting the tourism industry, which finds in these declarations the most effective brand image that a tourist destination can receive. The insertion of the heritage in this world or global scale transforms its nature: it will become part of the list of privileged places in the world; it becomes an icon of universal validity [24,25].

### 2.3. The Cemetery as Cultural Heritage

We also understand heritage—in its broadest sense—as a set of assets of economic or symbolic value. This term is linked to the notion of heritage, because it is a reflection of historical continuity. Similarly, the broader concept of "cultural heritage" goes beyond the purely historical or artistic. As pointed out before, UNESCO defines it as a cultural heritage that must be transmitted to future generations and, therefore, must be the object of study and protection. It is also a source of emotional experiences, due to its aesthetic, historical, ethnographic, or anthropological value.

The cemetery is a universe of symbols, reflected mainly in the architecture and sculpture it contains. It represents a replica of the city, that of the dead (necropolis), as opposed to that of the living (polis). In them we can observe different levels of representation. One is that of their location outside the walls, but they are also a reflection of social and economic status, manifested in a wide typology of pantheons, tombstones and tombs, niches and graves, and in their distribution inside the enclosure.

Thus, cemetery culture can be seen as an "immaterial landscape of memory" that accounts for cultural, social, and historical changes [26]. Cemetery culture tells us about funerary and mourning traditions in different epochs: what patterns developed within each society and how they related to it. Cemetery culture preserves biographies, mentalities, religions and beliefs, social structures, gender relations and, last but not least, local and regional particularities. Thus, cemetery culture can be interpreted as the history of a space that over time has been reinterpreted again and again according to the historical and social context. Therefore, cemeteries have become a kind of treasure trove of culture and society [12]. This is a heritage context that deserves to be studied, not only monographically in the case of a specific city or town, but also from a comparative perspective, in order to appreciate the similarities and differences in broad contexts.

The following section describes the context in which the cemetery is located, where this double experience is conducted: the development of thematic guides for visits and the proposal of a system for the classification of the heritage elements of the cemetery.

## 3. Materials and Method

### 3.1. Context

The history of the municipal cemetery of Murcia (Nuestro Padre Jesús) is the story of the modernization of a country. Its current planning, dating from the nineteenth century, shows how the cemetery was part of the new city, of the new services that were required of the consistory; it therefore had to be "monumental", as part of the theater or the public market, because the society had already relegated its initial prejudices and required the monumental in the funerary, both in the set and in the pantheons.

The configuration of the cemetery at the end of the 19th century opted for more orderly projects, which sought greater control of the results. To this end, the distribution of the pantheons was the pantheons flanking the roads that articulated the enclosure in the form of a Latin cross and around the central hill, which—because the chapel was not built—has remained as a picturesque trace foreign to the original idea. Although the building was originally to be built on a slope, which would have given it a more romantic layout, it was decided that they should flatten the terrain except for the hill. Figure 1 shows the layout of the cemetery:

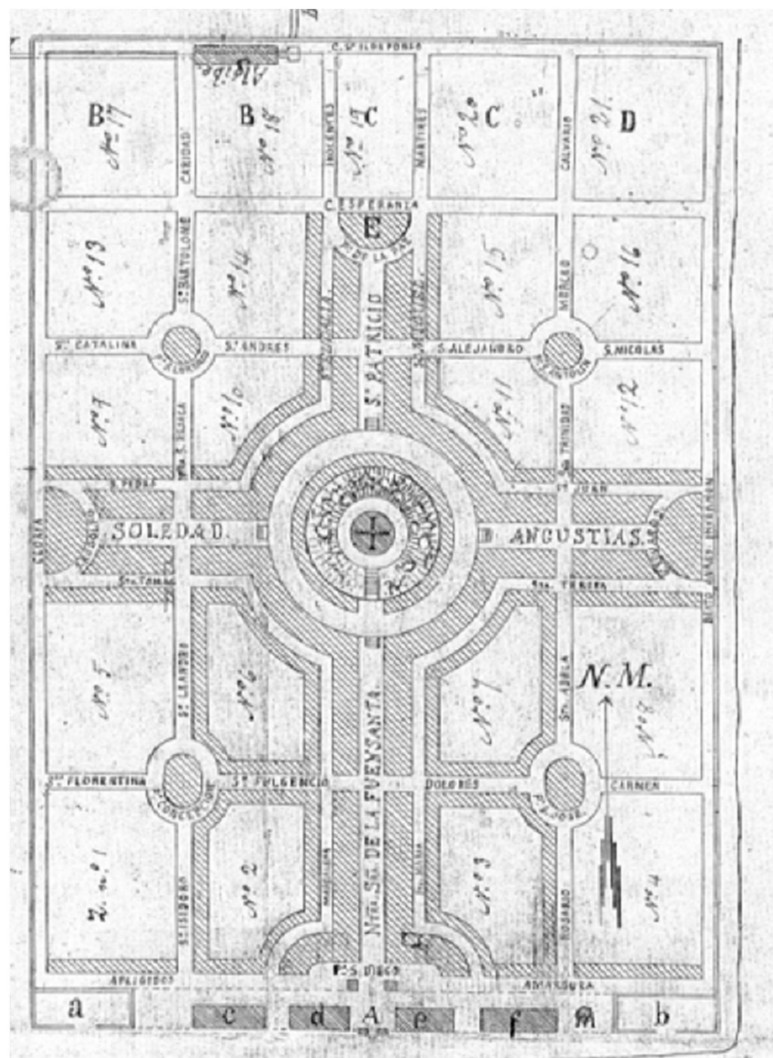

**Figure 1.** Plan of the spatial distribution of the original Cemetery "Nuestro Padre Jesús" in 1887.

It is proposed as a scheme of great monumentality that offers a hegemony of architecture over sculpture, both in pantheons and mausoleums, the latter being used in architectural decoration and in some specific works. If construction had begun earlier, it would have been more varied in terms of style, but the Murcian cemetery is dominated by eclecticism, as was the architecture of the city at the time the cemetery was built. It was maintained perhaps because of the symbolic components that funerary architecture required and that were associated with historical styles. The scarce presence of modernism is surprising, as is the case in the city, which is barely evident in one of the pantheons. Figure 2 shows an example of a pantheon we may find in the cemetery:

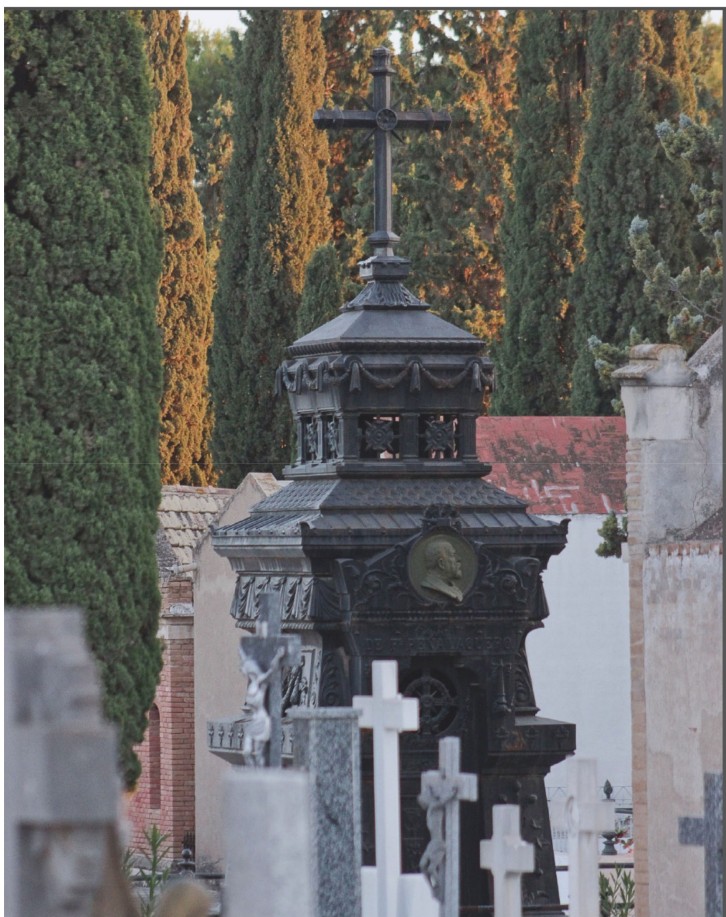

**Figure 2.** Example of a pantheon. Mausoleum of the entrepreneur Francisco Peña Vaquero (1836–1907).

*3.2. Methodology and Data Collection*

As exposed, this article shows the results of a research that has been conducted since 2015. It involves the development of different thematic guides to visit the Cemetery, also linked to the organization of guided tours that seek to revitalize the concept of funerary cultural heritage. The elaboration of the different guides was conducted with the collaboration of students of the Philosophy Degree of the University of Murcia as part of a teaching innovation project. Therefore, it is an experience developed in a public–private collaboration between three agents: the "Sociedad Murciana de Antropología" (SOMA), the City Council of Murcia, and the University of Murcia.

For the elaboration of the different visit guides, documentation tasks have been developed in public and private archives, as well as in-depth interviews to those key informants who have provided relevant data for the different biographies. For this task, different dimensions were elaborated to link the information of the people interviewed, and thus try to build the summaries for the teaching guides in a complementary way.

**4. Results and Discussion**

The history of the municipal cemetery of Murcia is the story of the modernization of a country and of how the City Council of the time, within its possibilities and overcoming the difficulties posed by the religious authorities, conducted this work. For the society of the 19th century, the cemetery was part of the new city, of the new services that were required of the consistory; it had to be "monumental", because, in the same manner as the theater or the public market, society had already relegated its initial prejudices and required the monumental in the funerary, both in the set and in the pantheons.

The municipal cemetery of the city of Murcia, located six kilometers north of the city center, was opened in 1887. It houses more than 165,000 tombs that tell the history of the

city itself and thus of Spain. In the same manner as all cemeteries, it is a mirror that reflects history, culture, and society. At the same time, it is a cultural landscape that is subject to a constant process of transformation. On the one hand, the original enclosure, characterized by a rectangular plan with an inscribed Latin cross, was enlarged in the 1950s, and on the other hand, the tombs have been removed from several areas and replaced by new graves. All these changes have been made without paying attention to issues of protection of the historical-cultural heritage and the enhancement of the wealth that houses the cemetery as a living book of history.

The focus on heritage issues began with the collaboration between the Murcian anthropological association SOMA, the University of Murcia and the Municipality of Murcia in an attempt to give visibility to the municipal cemetery as an asset of cultural interest. It is an effort between the three actors that has led to a whole series of activities and products that highlights the heritage of the cemetery.

### 4.1. A Three-Way Collaboration: Cemetery Tour Guides

In 2015 a collaboration agreement was signed between the Murcian Society of Anthropology (SOMA) and the City Council of Murcia, which aimed at promoting the visibility of the heritage of the cemetery. To this end, four lines of action were agreed: on the one hand the annual development of thematic guides for visits, on the other hand the offer of guided tours of the cemetery, then the organization of scientific meetings on funerary culture, and finally the publication of thematic issues of the Revista Murciana de Antropología, an anthropologic journal on funerary culture, published by SOMA. So, in 2015, a first introductory guide was published, which on the one hand develops the concept of the cemetery as a cultural property and on the other hand presents the history of the municipal cemetery in question.

The following thematic guides have been elaborated since then, in a subject of Social Anthropology that is offered at the University of Murcia and is developed in the form of student research projects following the methodology "research based-learning" [27,28]. In each of the thematic guides, twelve stories of people or families buried in the municipal cemetery are presented under the theme of the year. In this sense, we find a guide dedicated to "Writers and artists" (Figure 3a, 2016), "Elites of the nineteenth century" (Figure 3b, 2017), "Murcian women in memory" (Figure 3c, 2018), "Protagonists of progress" (Figure 3d, 2019), "Other Murcian. The traces of Europeanization" (Figure 3e, 2020), and "The 1918 flu epidemic in Murcia" (Figure 3f, 2021).

The second axis of the enhancement are the visits to the cemetery that during the two years (2017 and 2018) have been offered in the winter months; namely, fifteen guided tours to the cemetery. The visits were conducted as a free offer and, according to legal requirements, were conducted by official tour guides, taking into account the guidelines and material prepared by the research team. Another activity that aids in completing the efforts is the organization of scientific meetings, which is open to the public. A first meeting was held in 2018 and a second in 2020; always with the participation of international specialists. It is the contributions of these specialists that have subsequently been used for the elaboration of two thematic issues of the Revista Murciana de Antropología.

A total of two international congresses have been held with the intention of promoting an exchange that would allow a comparative study of "funerary heritage cultures". In 2018, different European specialists were brought together at the International Symposium "Funerary Cultures in Europe". In 2020, with the intention of sharing the results of new experiences, the International Congress, "The cemetery as a place of European memory", was held.

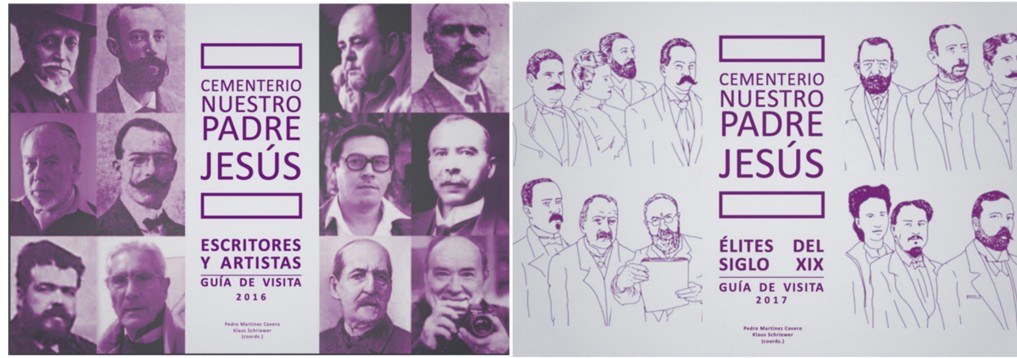

(**a**) Guide "Writers and artists"          (**b**) Guide "Elites of the 19th Century"

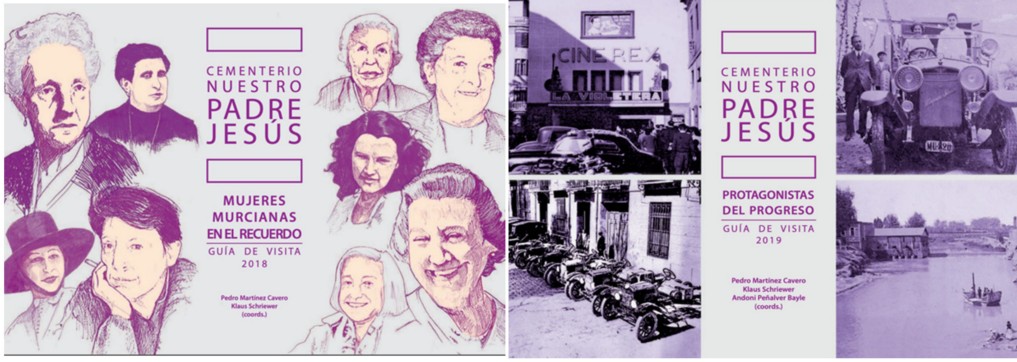

(**c**) Guide "Murcian women in memory"          (**d**) Guide "Protagonists of progress"

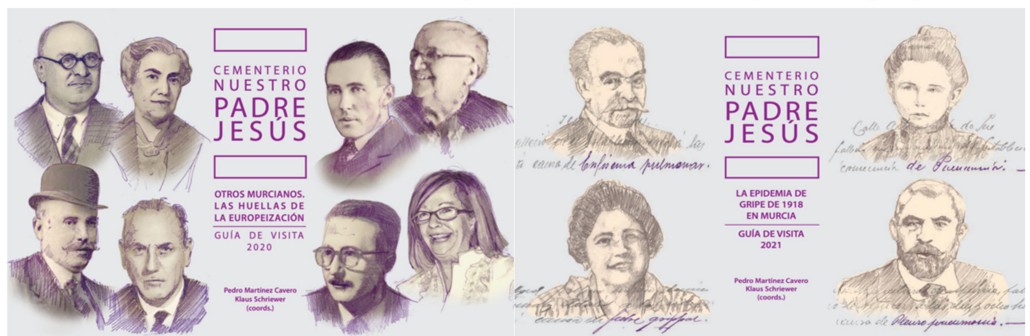

(**e**) Guide "Other Murcian. The traces of Europeanization"          (**f**) Guide "The 1918 flu epidemic in Murcia"

**Figure 3.** Cover of the six thematic guides to visit the Cemetery.

*4.2. The Municipal Cemetery as an Asset of Cultural Interest—Challenges behind a Necessary Change*

Since the beginning of the project, there has been a debate on the need to apply the category of Object or Asset of Cultural Interest to the site and thus formalize the obligation to protect it. This is evident in certain areas of the cemetery, as in the case of zone 18, which once housed children's graves from the post-war period, thus documenting the high infant mortality of the so-called hunger years and the sanitary misery of the time. The area with its children's graves was the only testimony in Murcia that reflected the difficulties of the post-war period and therefore had a high conservation value, although aesthetically it was not striking. However, these tombs were removed from the cemetery in 2008. This example shows the difficulty and in some respects the dilemma that occurs in this as in any other cemetery: which of the graves and installations deserve to be preserved and protected? How can one decide which of the elements of a cemetery should or should not be preserved?

This question has gained new topicality with the request that the association Huermur, which works for the defense of the cultural heritage of Murcia, has submitted to the city council of Murcia and that forces the administration to declare the cemetery in its original extension, as a Property of Cultural Interest. This opens a debate on, more generally speaking, the criteria to be adapted for the valuation of a cultural landscape in constant transformation as the cemetery.

Within the framework of the collaboration with the city council, the research team presented a proposal for action. It is inspired by the concept developed by a group of Danish anthropologists with whom we have a close collaboration. This concept focuses on maritime heritage and in particular on traditional boats in Denmark. The principles were published in the so-called Copenhagen Charter on the Preservation of Ships [29], which presents the scientific features of the new method the Ship Preservation Fund had with the aim of creating greater rigor and appropriateness in the Foundation's assessment of the conservation value of ships.

It is interesting to note the multiplicity of versions and "voices" that, on many occasions, make it difficult to reach a consensus on the conservation of the heritage elements of the cemetery. In this case, a clear example of the often-conflicting interpretations of history is the Cemetery "Nuestro Padre Jesús", in which are the graves of victims of the Civil War. In Figure 4, we find a monument to the "Fallen for Freedom", honoring some 200 people executed in the first years after the Civil War (Figure 4a), between 1939 and 1943, and a monument to more than 120 soldiers of the International Brigades (Figure 4b).

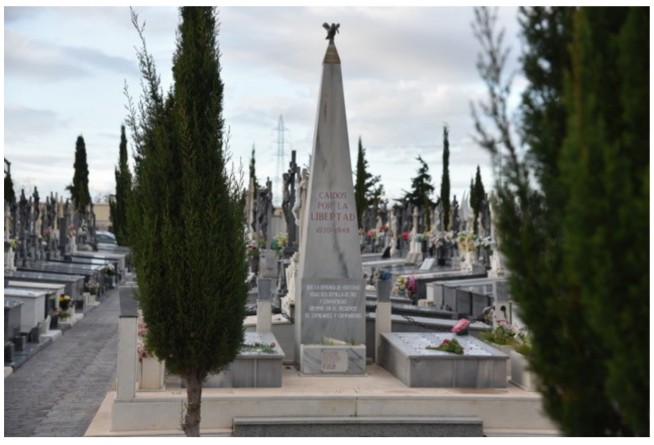

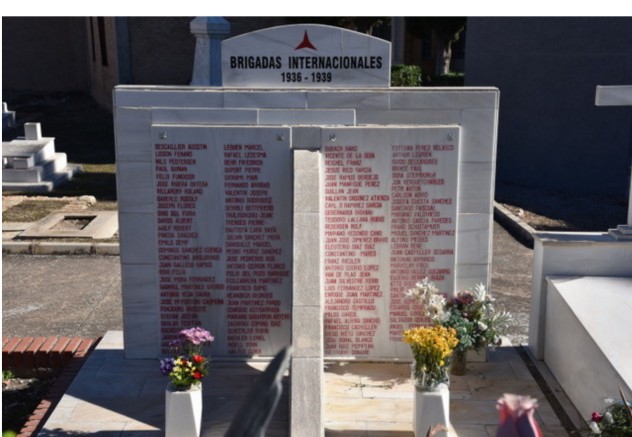

(**a**) Monument "Fallen from Freedom"                    (**b**) Monument dedicated to "International Brigades"

**Figure 4.** Examples of memorials to the Republican victims of the Civil War.

On the other hand, Figure 5 shows the grave of a victim linked to the right-wing side who finally won the war, giving rise to a 40-year dictatorship in Spain. We can see in the example of this tomb how the narratives about the history of the war, in this case, change depending on how these tombs are interpreted. In this last tomb, in Figure 5, we can read "fallen for God and for Spain. He was killed by the Frente Popular".

*4.3. A Proposal for The Elaboration of a BIC Register in Nuestro Padre Jesús*

The conservation activities that the City Council of Murcia has begun in the Cemetery Nuestro Padre Jesús in recent times are a first step towards the protection of the heritage elements gathered here as witnesses of history. Among the thousands of graves, there are several that are undoubtedly of great interest, because they have a great artistic value, allow a greater understanding of the past, and can be a vehicle that allows citizens to reflect on identity and belonging. However, there are many graves that at first glance do not stand out. In this case, we should decide criteria to apply in order to decide which graves deserve attention, beyond the maintenance efforts that families want or can devote to them. That is to say, in this case of having a patrimonial consideration, an adequate maintenance

should be conducted according to this criterion, beyond the maintenance given by the families privately.

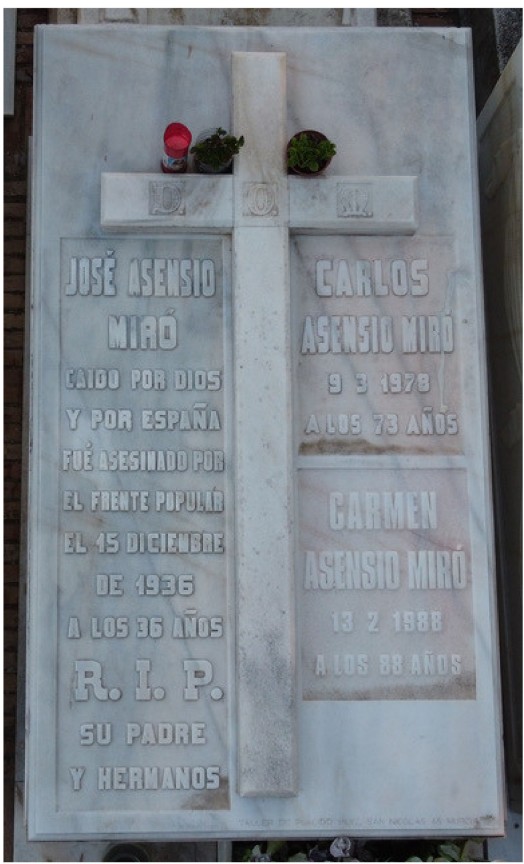

**Figure 5.** Tomb with inscriptions referring to his participation in the Civil War.

Taking advantage of the philosophy of the procedure successfully applied in Denmark [29], it is possible to elaborate a system that, adapted to the cultural landscape of cemeteries, takes into account three dimensions that can be linked to each object: the general, the particular, and the individual. The basic principles of these dimensions are as follows.

*4.4. General Dimension*

The general dimension ascertains the generic features of a cemetery, of the graves, and of other objects of remembrance found in the enclosure. Applying this general perspective to the Cemetery, we can differentiate between Muslim, Catholic, and Protestant graves as well as graves that do not show religious ties. On the other hand, we must differentiate between individual graves, family graves, and collective graves.

This means that if one of these general characteristics is in the process of disappearing, a grave with this feature may have a high general conservation value, although it does not have a high special conservation value. For example, it may be that in a cemetery the graves of suicides disappear and a few remain, and it would be of interest to keep them even if they have a limited individual value.

*4.5. Particular Dimension*

The particular dimension refers to the different types of graves, such as pantheon, mausoleum, vault, or raised tomb, etc. It defines and locates the different types of graves in their shape and historical period in order to elaborate a catalog of the different types that exist in the cemetery and thus provide a tool to select graves that represent the different particular types and architectural styles, without necessarily standing out.

*4.6. Individual Dimension*

The individual dimension represents the outstanding characteristics of the singular grave that derive from its architecture, artistic value, socio-cultural importance, historical informative value, conservation, and even related documentation. In cemeteries, there are a number of graves that stand out in this regard. There are some graves that are unique in their architecture, have a great informative value with respect to the history of the city, are graves of prominent people, or even combine several of these criteria. Thus, we can highlight, for example, the cast iron pantheon of the industrialist Francisco Peña Vaquero or the sarcophagus of the Pagán Ayuso family (Figure 6), whose conservation in its current state should have a high priority.

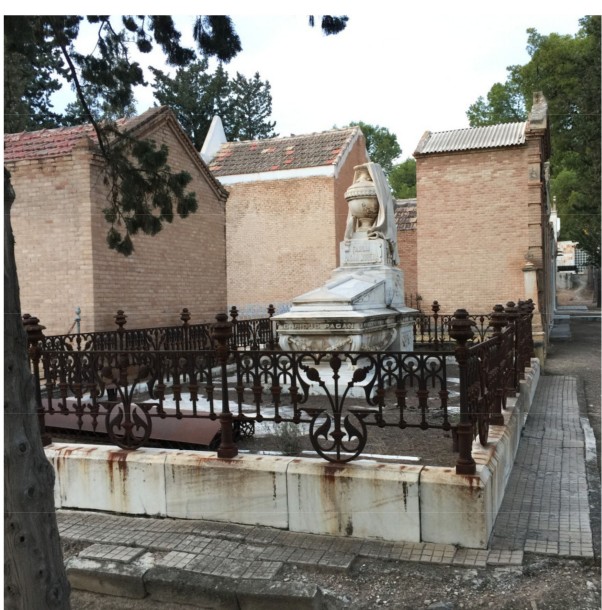

**Figure 6.** Sarcophagus of the Pagan Ayuso family in the cemetery.

Taking into account these three dimensions, a catalog of criteria can be established that allows a list of assets to be created according to their heritage significance. Table 2 proposes this analysis in relation to the dimensions we propose for the case of the Cemetery.

**Table 2.** List of dimensions and categories to analyze the elements of the cemetery.

| Dimension | Categories |
|---|---|
| General<br>It specifies the essential features that, in a generic way, characterize the graves and allow a first differentiation. | Catholic burials.<br>Protestant graves.<br>Muslim graves.<br>Non-religious graves.<br>Suicide graves.<br>Individual graves.<br>Family burials.<br>Mass graves.<br>Graves of other groups. |
| Particular<br>It specifies the cultural traits that characterize specific types of burials, their use, and development. | Columbarium.<br>Niche.<br>Tomb.<br>Raised tomb.<br>Vault.<br>Pantheon.<br>Mausoleum.<br>Monument. |

Table 2. *Cont.*

| Dimension | Categories |
|---|---|
| **Individual**<br>It specifies the cultural traits that characterize architecture, use, and change. | Criteria to be considered:<br>Scope of the relative documentation of the burial.<br>Individual narrative value of the burial.<br>Owner of the grave: importance in Murcian society.<br>Builder and architect of the grave.<br>Purpose of the burial.<br>Craftsmanship: special refinement or specially designed artisan details.<br>Important historical relationships.<br>Aspects of maintenance or modification of the tomb.<br>Rarity of the grave.<br>Aesthetic qualities of the grave.<br>Specific details of the grave (ornaments etc.). |

## 5. Conclusions

Introductory manuals on cultural heritage usually define it as the body of knowledge that is transmitted between generations in order to refer to the past and understand the present. They usually highlight its invaluable task of giving identity and guaranteeing group cohesion. Once these basic pieces have been offered to initiate us in the understanding of what cultural heritage embraces, they usually organize the chapters of the book by "watertight drawers", classifying heritage elements in a hermetic way: festivals, gastronomy, oral tradition, and museums. On some occasions, they refer to the idea of the social construction of heritage: heritage does not exist in nature, and is given by a social and cultural hegemony in a particular place and time. Less often they talk about how heritage is managed [30]. This article has attempted to review this concept of cultural heritage, specifically in relation to the complicated but necessary task of determining how to reach a consensus on what is considered cultural heritage. For this purpose, the case of the heritage process of the Nuestro Padre Jesús Cemetery in Murcia (Spain) has been presented.

How should cultural heritage be understood in cemeteries in the 21st century? This question, which deserves an answer in several levels, allowed us to speculate on the existence of at least two ways of "doing" cultural heritage. First, there will be a heritage to which we turn to meet our affections: the cuisine of my grandparents, the music of my people, the parades linked to religious celebrations. The "current folklores", now devoid of political intention or interest in social change, reproduce ad infinitum these friendly scenarios. Second, beyond these representations (more psychological than social) there must be another heritage (or heritages) that acts as a "major discipline", with analytical capacity and willingness to intervene in contemporary issues of interest. This second dimension is the one that leads us to propose this experience where cultural heritage occupies its place of memory and identity, going beyond the descriptive level, and also acts as a vehicle to offer an experience of teaching innovation [31].

The Cemetery Nuestro Padre Jesús stands as a singular enclosure of the city, a witness of time, and a testimony of its cultural diversity. A place of memory, a monumental and symbolic space which allows a cultural reading, because it has been the place erected and instituted to represent memory and is a part of the social rituals surrounding death. It is a cultural heritage that reflects the lights and shadows of our history and offers an account of social and religious life. The cemetery allows the construction of regional history, because it is the resting place of characters recognized by the "official history" of the region of Murcia.

It is interesting to note the multiplicity of versions and "voices" that, on many occasions, make it difficult to reach a consensus on the conservation of the heritage elements of the cemetery. This tension between the traditional and new patterns has its reflection in the funerary world and is materialized in the cemetery, where we observe more and more 'bricolage' between the known and alternative forms. The transformations will depend on a multitude of interrelated factors. First of all, we should mention the changes in beliefs

and progressive secularization. Second, we refer to the diversification of family models, with the corresponding perception of the responsibilities and roles that their members can develop. Other factors that should be taken into account are the discourse on public health, the increase in populations and their growing mobility, the economic difficulties of municipalities and the legislative developments in the area of public health.

In addition to all these changes, in recent years there has been a growing interest in the cemetery as a cultural heritage and witness to history. This is why we believe that the history of a cemetery documents—albeit in a specific way—the socio-cultural transformations and the corresponding history of a place. The characteristics and changes of a society are materialized in the very structure of the cemetery itself, in the way the tombs are distributed on the site, in the different shapes of the tombs, and in the symbolic language, etc. Given this agency from the contemporary cemetery, concrete proposals for the management of its heritage elements are necessary. The experience presented in this article demonstrates the possibilities of active management of the cemetery and, therefore, the necessary enhancement of its heritage elements in a planned manner.

Finally, we must highlight the importance of carrying out, in parallel to this proposal, a process of digitization of the information and assets linked to the cemetery. This would be a large-scale process that would involve collaborative work between the administration, the university, and civil society, as well as the intervention of multidisciplinary experts who contribute value from different disciplines of knowledge. In other words, it is an aesthetic, historical, and identity work, as well as a task of analysis of materials, architectural and sculptural techniques, and urban planning of the layout of the cemetery itself.

**Author Contributions:** Conceptualization, G.L.-M. and K.S.; methodology, G.L.-M and K.S.; software, G.L.-M. and K.S.; validation, G.L.-M. and K.S.; formal analysis, G.L.-M. and K.S.; investigation, G.L.-M. and K.S.; resources, G.L.-M. and K.S.; data curation, G.L.-M. and K.S.; writing—original draft preparation, G.L.-M. and K.S.; writing—review and editing, G.L.-M. and K.S. All authors have read and agreed to the published version of the manuscript.

**Funding:** The activities of the Jean Monnet Chair "Cultural aspects of European integration" (EACEA, Project number 610698-EPP-1-2019-1-ES-EPPJMO-CHAIR).

**Institutional Review Board Statement:** Not applicable.

**Informed Consent Statement:** Not applicable.

**Acknowledgments:** We want to thank the support received from Eduardo Gonzalez Martínez-Lacuesta, who from the beginning believed in this project in the Nuestro Padre Jesús Cemetery.

**Conflicts of Interest:** The authors declare no conflict of interest.

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
