# Peer review of "Challenges in the Valorization of the Funerary Heritage; Experiences in the Municipal Cemetery of Murcia (Spain)"

_heritage, doi:10.3390/heritage5010007_

Round 1
Reviewer 1 Report
Dear Authors
I appreciate your manuscript; it is interesting and well organised.
My only comment concerns the references that do not in all cases follow the journal requirements.
I could moreover suggest that could be interesting to couple this research with a petrographical mineralogical characterization of the natural stones used in the cemetery.
Best regards
Author Response
Dear reviewer,
Thank you very much for your comments.
Regarding the references, we have followed the indications of the template that can be found in the journal itself:
"References must be numbered in order of appearance in the text (including citations in tables and legends) and listed individually at the end of the manuscript. We recommend preparing the references with a bibliography software package, such as EndNote, ReferenceManager or Zotero to avoid typing mistakes and duplicated references. Include the digital object identifier (DOI) for all references where available".
In this case, we have used Mendeley Desktop as the reference manager. So they are "numbered in order of appearance in the text (including citations in tables and legends) and listed individually at the end of the manuscript". We have used the same manager in other publications and it has been considered successful.
On your second proposal. Of course, such an analysis, which also takes into account the materials, would be very interesting. In this sense, we have included a final paragraph in the conclusions section, specifying this proposal for the future. It would be a collaboration between different disciplines that would enrich this type of work.
Kind regards
Reviewer 2 Report
The article concentrates on the exploration of prioritization/evaluation of the cultural assets in CNJP in Murcia (Spain). The necessity for this evaluation process was a natural result of the collaborative project for what are the assets and values of cultural nature to be protected in CNJP. The project(s) has collected data over six years in the cemetery and six thematic visit guides were prepared and dissemination of the results were achieved by guided tours and scientific meetings including international congresses open to the public.
The results have been represented well. However, there is a need for explanaton for the issues below:
- How the data gathered will be stored and disseminated for the whole cemetery? For example is there any plan to use digital technologies for the storage and or dissemination?
- In addition to aesthetical, formal and historical data collected is there any level of documentation of the material use?
- What is the state of conservation of the cemetery and in the cemetery?
- If there is an evaluation of the state of the conservation at any level how would that effect the value of the assets in the cemetery?
Thank you vey much and please find the attached document…
Best

Author Response
Dear Reviewer,
Thank you so much for your comments. It contributes to improve our proposal.
Please, find attached our reply.
Best regards

Round 2
Reviewer 2 Report
Thank you very much for the corrections and your notes. I think the manuscript is good to be published. Also please note that lines 176-178 was highlighted since it gives the importance of the subject (i apologize if it created any misunderstanding).
Best,
K. G. Akoglu